

# Calcareous dinoflagellate blooms during the Late Cretaceous 'greenhouse' world—a case study from western Ukraine

Agnieszka Ciurej[1], Zofia Dubicka[2,3] and Andriy Poberezhskyy[4]

[1] Institute of Biology and Earth Sciences, Department of Geology and Palaeontology, Pedagogical University of Krakow, Kraków, Poland
[2] Faculty of Geology, University of Warsaw, Warsaw, Poland
[3] GFZ German Research Centre for Geosciences, Telegrafenberg, Potsdam, Germany
[4] Institute of Geology and Geochemistry of Combustible Minerals of NAS of Ukraine, Lviv, Ukraine

## ABSTRACT

The Late Cretaceous was a unique period in the history of the Earth characterized by elevated sea levels, reduced land area, and significantly high concentrations of atmospheric $CO_2$ resulting in increased temperatures across the globe—a 'Greenhouse World'. During this period, calcareous dinoflagellate cysts (c-dinocysts) flourished and became a ubiquitous constituent of calcifying plankton around the world. An acme in calcareous dinocysts during the Albian to the Turonian coincided with the highest recorded seawater surface temperatures and was possibly linked to conditions that favored calcification and a highly oligotrophic system in European shelf seas. This study examines the potential applicability of c-dinocysts as a proxy for paleoenvironmental conditions based on their assemblage changes plotted against foraminiferal occurrences and microfacies analysis. The material was extracted from the upper Turonian chalk of the Dubivtsi region in western Ukraine. An inverse correlation was observed between species diversity and the number of c-dinocyst specimens. Nutrient availability gradients apparently determined important changes in the calcareous dinocysts distribution. These trophic changes were likely caused by the interplay of eustatic sea-level fluctuations and Subhercynian tectonic activity leading to changeable nutrient inputs from the nearby land.

## INTRODUCTION

Calcareous dinoflagellate cysts, also known as calcareous dinocysts (c-dinocysts), are minute calcite structures measuring between 10 and 180 µm; they are produced by peridinoid dinoflagellates belonfging to the family Thoracosphaeraceae, which is classified under the Order Peridiniales, Class Dinophyceae (*Elbrächter et al., 2008*). Dinoflagellates that produce calcareous cysts are photosynthetic planktonic organisms (*Tangen et al., 1982*; *Montresor et al., 1994*). Modern c-dinocysts, which are primarily dominated by *Thoracosphaera heimii*, are present in many regions around the world (*Dale, 1992*; *Montresor, Zingone & Sarno, 1998*; *Höll et al., 1999*; *Vink, Zonneveld & Willems, 2000*). Approximately 30 extant species

Corresponding author
Agnieszka Ciurej,
agnieszka.ciurej@up.krakow.pl

and about 260 fossil species (morphotypes) (*Nowak, 1963*; *Reháková, 2000*; *Fensome & Williams, 2004*; *Ciurej & Bąk, 2021*) are known to date.

The earliest fossil record of c-dinocysts is from the Upper Triassic of the Alps (*Janofske, 1992*). They are widely used as a marker in biostratigraphy, paleoclimate, and paleoenvironmental reconstructions (*Keupp, 1991*; *Zügel, 1994*; *Dias-Brito, 2000*; *Wendler, Gräfe & Willems, 2002*; *Wendler, Wendler & Huber, 2013*; *Omaña et al., 2014*; *Wiese, Zobel & Keupp, 2015*; *Ciurej, Bąk & Bąk, 2017*). During the Late Cretaceous, c-dinocysts were distributed world-wide (*Keupp, 1991*; *Willems, 1994*), with *Pithonella* often dominating the assemblages (*Dias-Brito, 2000*; *Wendler, Gräfe & Willems, 2002*). Pithonellids have been recorded from both the northern and southern hemispheres, including Western Europe (*Wendler, Gräfe & Willems, 2002*; *Wiese, Zobel & Keupp, 2015*), Central Europe (*Nowak, 1963*; *Reháková, 2000*; *Ciurej, Bąk & Bąk, 2017*), Tadzhikistan (*Andryuschenko & Dolitskaya, 1975*), the South Atlantic Ocean (*Dias-Brito, 2000*), and Middle America (*Omaña et al., 2014*). The widespread acme of pithonellids during the Albian–Turonian coincided with maximum surface-water temperatures and the mid-Cretaceous transgression that led to expansion of relatively shallow and warm epicontinental seas (*Hallam, 1992*; *Miller, Wright & Browning, 2005*; *Gale et al., 2008*). One of the reasons for the pithonellid bloom during this period may have been the favorable conditions of seawater that promoted calcification, including a high calcium-ion concentration and low Mg/Ca ratio (*Stanley, Ries & Hardie, 2005*; *Van Dijk et al., 2016*) together with a high concentration of dissolved inorganic carbon (DIC) (*Kelly, 2003*) which is a limiting compound for $CaCO_3$ precipitation in sea-water today (*Erez, 2003*). The *Pithonella* acme might be also related to the widespread oligotrophic conditions in the early Late Cretaceous shelf seas (*e.g.*, *Wiese, Zobel & Keupp, 2015*). Pithonellids are interpreted to be opportunistic organisms that can thrive, evolve, and bloom in oligotrophic-mesotrophic environments and were also able to adapt to eutrophic environments (*Dias-Brito, 2000*). Paratabulation (openings—archeopyles and covering plates—opercula) have been recently documented in pithonellids (*Leighton, Hart & Smart, 2011*; *Wendler, Wendler & Huber, 2013*; *Wendler & Bown, 2013*). This finding confirms dinoflagellate affinity of pithonellids. Modern dinoflagellates can overcome temporary nutrient deficiency by producing resting cysts (*e.g.*, *Vink, Zonneveld & Willems, 2000*). Abundant but low-diversity c-dinocyst assemblages have been documented from surface sediments in oligotrophic areas of the Atlantic Ocean (*Höll et al., 1999*; *Vink, Zonneveld & Willems, 2000*; *Zonneveld, Brune & Willems, 2000*; *Zonneveld et al., 2005*).

The upper Turonian pure white chalk of the Ukrainian portion of the European epicontinental sea (southeastern part of the Central European Basin), exhibits abundant c-dinocysts (*Bojczuk & Woloszina, 1971*; *Rozumeyko, 1978*; *Pasternak, 1984*; *Olszewska et al., 2012*; *Dubicka & Peryt, 2012*). In some intervals, c-dinocysts are the main constituents of the chalk deposits. In addition, the strong dominance of one species, *Pithonella ovalis* (Kaufmann in Heer 1865) Lorenz 1902 indicates single-species blooms. Therefore, during the relatively long period of the late Turonian (91.31–89.75 Ma; *Laurin et al., 2014*; *Plint et al., 2022*), the surface waters of this basin were largely inhabited by these organisms. However, c-dinocysts have not been described in detail in the study area. The published
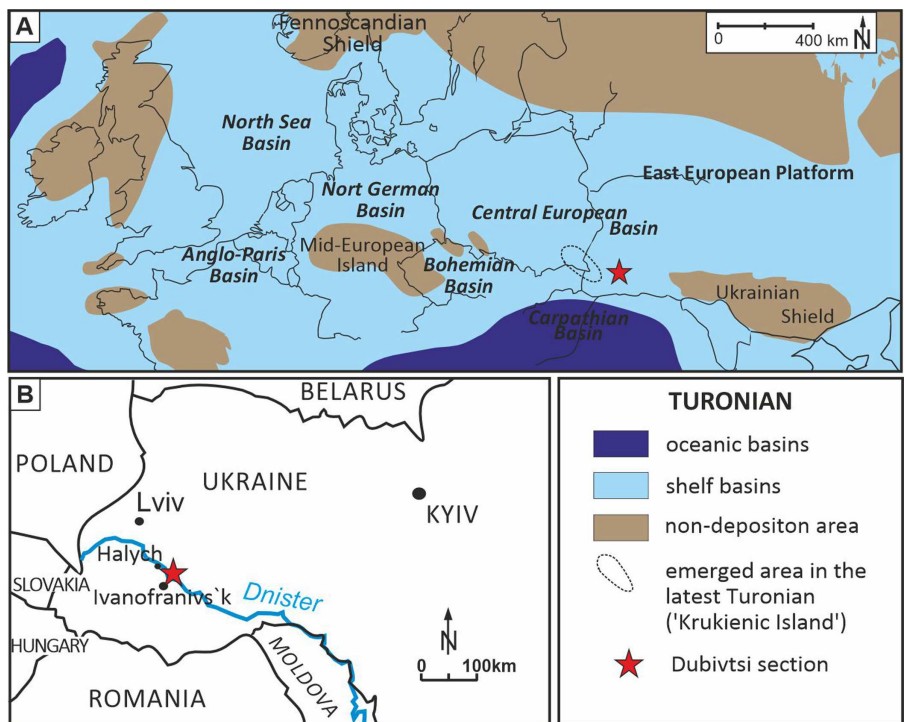

**Figure 1** **Location of the Dubivtsi section.** (A) Paleogeographical map during Turonian time (∼90 Ma), (modified after *Pasternak, 1959*; *Janetschke & Wilmsen, 2014*; *Scotese, 2014*). (B) Sketch map with location of the studied Dubivtsi section, (modified after *Dubicka & Peryt, 2012*).

data refer only to general information that mostly mentions their presence. In this study, we present a detailed qualitative and quantitative analysis of the c-dinocysts of the upper Turonian chalk of the Dubivtsi section plotted against microfacies and foraminiferal assemblage composition (*Dubicka & Peryt, 2012*). These data help to constrain the environmental factors that caused c-dinocysts bloom in the Late Cretaceous.

## GEOLOGICAL SETTING

The Dubivtsi section is exposed in a quarry located 1 km east of Dubivtsi, near Halych (Fig. 1). The section comprises four lithological units from the bottom to the top (*Dubicka & Peryt, 2012*): (1) white chalk (25 m thick, strongly lithified, with $CaCO_3$ content ranging from 97.8 to 99.9%); (2) hardground; (3) inoceramid limestones (4.4 m thick); and (4) marls (1.5 m thick). The total thickness of the studied section is 31 m. The c-dinocysts were studied from the white chalk interval.

A late Turonian age of the white chalk in the Dubivtsi section is indicated by planktonic foraminifera that are characteristic of the uppermost part of the *Helvetoglobotruncana helvetica* and *Marginotruncana coronata* local foraminiferal zones (*Walaszczyk & Peryt, 1998*; *Peryt, Dubicka & Wierny, 2022*); the *Marginotruncana coronata* Zone of the Central European Basin (*Walaszczyk & Peryt, 1998*; *Peryt, Dubicka & Wierny, 2022*) correlates to the *Inoceramus costellatus* (*I. perplexus,* see *Wiese, Čech & Walaszczyk I. Košták, 2020*) and

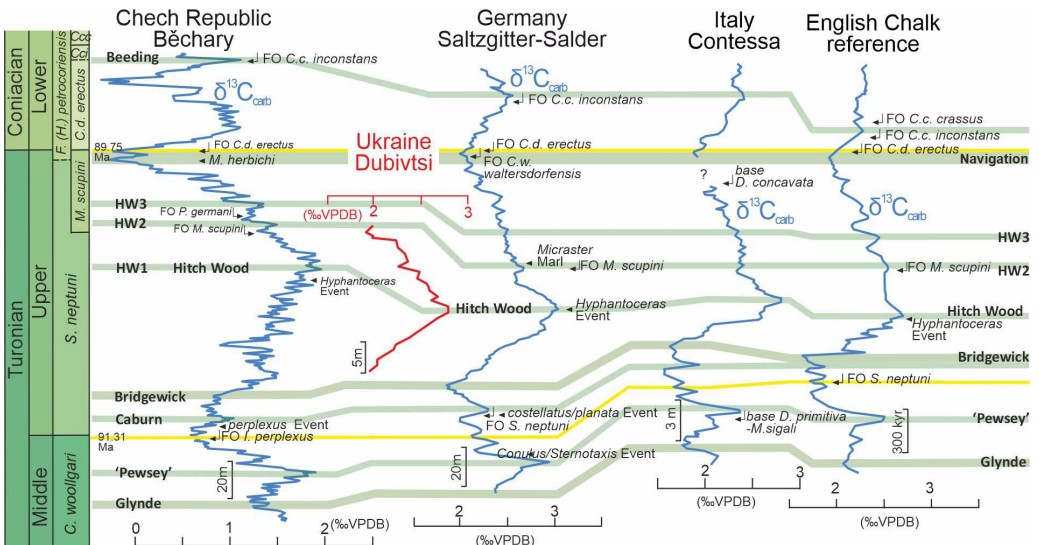

**Figure 2** Chemostratigraphic correlation of the Dubivtsi section with European Turonian sections using bulk-carbonate carbon stable isotopes. Correlation between references profiles after *Jarvis et al. (2015)*. Isotope data source: Běchary—*Jarvis et al. (2015)*; Saltzgitter-Salder—*Voigt & Hilbrecht (1997)*; Contessa—*Stoll & Schrag (2000)*; English Chalk—*Jarvis et al. (2006)*. Correlation of positive (green) and negative (red) carbon isotope excursions defining named carbon isotope events after *Jarvis et al. (2006)* and *Jarvis et al. (2015)*.

*Mytiloides scupini* inoceramid zones. In addition, the co-occurrence of *Marginotruncana coronata* (Bolli, 1945), *Marginotruncana marginata* (Reuss, 1845), *Marginotruncana pseudolinneiana* Pessagno 1967, *Dicarinella imbricata* (Mornod, 1950) and representatives of *Falsotruncana* indicates the upper Turonian (*Huber et al., 2017*; *Haynes et al., 2017*) *Marginotruncana schneegansi* standard planktonic foraminiferal zone (*Gradstein et al., 2020*). The last occurrence of *H. helvetica* in the Boreal Province (*Walaszczyk & Peryt, 1998*) seems to be slightly higher than in the Tethyan and Austral realms (*Huber & Petrizzo, 2014*). A late Turonian age for the Dubivtsi chalk (below the hardground) is supported by the benthic foraminiferal assemblage: *Gavelinella ammonoides* (Reuss, 1844), *Gavelinella vesca* (Bykova, 1939), *Cibicides polyrraphes* (Reuss, 1845), *Globorotalites multiseptus* (Brotzen, 1936) (see *Walaszczyk et al., 2022*; *Strózyk et al., 2018*). The broad peak displayed by the $\delta^{13}$C carbonate profile from the Dubivtsi section (*Dubicka & Peryt, 2012*) is characteristic of the upper Turonian throughout Europe (*Gale, 1996*; *Jarvis et al., 2006*; *Jarvis et al., 2015*) with the positive excursion at the peak maximum representing the Hitch Wood Event, a global marker (*e.g.*, *Plint et al., 2022*) (Fig. 2).

Geologically, the study region is situated on the southwestern margin of the East European Platform within the Lviv-Stryi Syncline (*Pasternak et al., 1968*; *Pasternak, Sen'kovskyi & Gavrylyshyn, 1987*; *Świdrowska et al., 2008*), which is the southernmost part of a larger tectonic unit called the Border Synclinorium. The Border Synclinorium extends to the northeastern margin of the Mid-Polish Anticlinorium (*Pozaryski et al., 1979*). The Turonian deposits of the Dubivtsi section belong to the lithological unit here referred

to as the Dubivtsi Formation and originally named the "Dubivtsi Suite" (*Pasternak, 1959*; *Ivannikov et al., 1987*; *Gavrilishin, Pasternak & Rozumeyko, 1991*), which extends over a large area from Volhynia to Podolia in western Ukraine and overlies Cenomanian limestones. The Dubivtsi Formation ranges in thickness from 54 to 158 m and is divided into two distinct intervals. The lower interval, which is of Turonian age, comprises white and gray limestones that are rich in *Pithonella*, as identified by *Rozumeyko (1978)*. It also contains flint concretions that appear 15–20 m above the base of the formation and become more abundant toward the top of the section (*Gavrilishin, Pasternak & Rozumeyko, 1991*; *Vashchenko et al., 2007*). The upper interval, which is of Coniacian age, consists of limestones, argillaceous limestones and marls that often contain fragments of inoceramids (*Gavrilishin, Pasternak & Rozumeyko, 1991*).

The Upper Cretaceous strata of the Dubivtsi section were deposited in the southeastern part of the Cretaceous Central European Basin, c.a. 100 km south of the Carpathian Basin (*Ziegler, 1990*, and Fig. 1A). During the late Turonian, the Dubivtsi area was submerged under an epicontinental sea (Fig. 1A) with variable water depth from c.a. 50 m up to slightly above 100 m (*Dubicka & Peryt, 2012*) interpreted based on planktonic foraminiferal morphogroups (see *Bé, 1977*; *Hart & Bailey, 1979*; *Leckie, 1987*). During most of the late Turonian, Dubivtsi was far away from emergent areas; the nearest land (uplifted Ukrainian Shield) was at least 300 km to the east (Fig. 1A). An uplift of the 'Krukienic Island' (*Pasternak, 1959*) (ringed area in Fig. 1A) located around 50 km NW of the Dubivtsi began possibly during the latter part of the late Turonian (*Pasternak, Sen'kovskyi & Gavrylyshyn, 1987*; *Dubicka, Peryt & Szuszkiewicz, 2014*) (Fig. 1B). A previous study of the benthic foraminiferal assemblages (*Dubicka & Peryt, 2012*) showed that sedimentation of the Dubivtsi chalk occurred under oligotrophic conditions, as reflected by the low species diversity and extremely low abundance of planktonic and benthic foraminifera and a flood of pithonellids.

## MATERIAL AND METHODS

Sixteen samples of white chalk from the Dubivtsi section were analyzed for c-dinocysts. These samples were a subset of the 32 samples that were previously studied for foraminifera (*Dubicka & Peryt, 2012*) (Fig. 3). Sixteen rock thin sections 3 × 5 cm in size were prepared for assessing c-dinocyst abundance and diversity as well as microfacies analysis. To estimate the total c-dinocyst content, the thin sections were arbitrarily divided into three transects within the lower, middle, and upper portions of thin section. Each transect was 0.47 mm wide and 19.6 mm long. The method used to quantify the c-dinocysts involved counting the number of species present in each lamina, which had an area of 27 mm$^2$ for each sample. The counts for each species were then summed to obtain the total content of each species of c-dinocysts in each sample. The same thin sections were used for microfacies analysis. Observations were made using a Nikon Eclipse LV100N POL polarizing optical microscope with a digital camera and NIS-Elements BR software for capturing photographs of thin sections (Department of Geology and Paleontology, Pedagogical University of Krakow).

Additionally, all sixteen samples were disintegrated for c-dinocyst studies, using the liquid nitrogen method (*Remin, Dubicka & Kozlowska A. Kuchta, 2011*). The residues

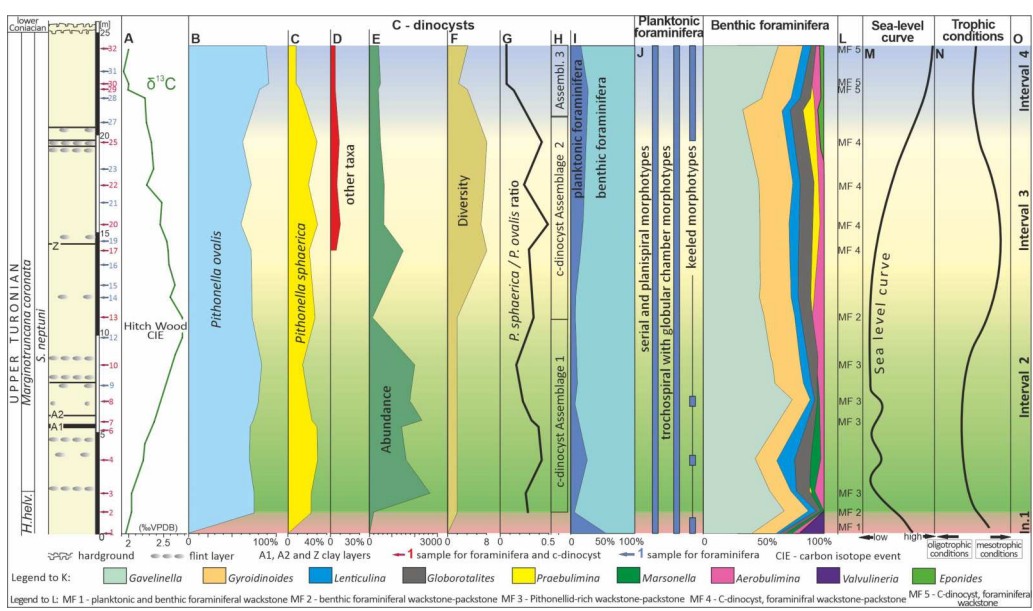

**Figure 3 Microfossil assemblage changes within the upper Turonian chalk of the Dubivtsi section.**
(A) $\delta^{13}C_{carb}$ curve for Dubivtsi chalk. (B–D) Percentage content of c-dinocysts (calculated in thin section).
(E) Total number of c-dinocysts (calculated in thin section). (F) Sample c-dinocysts species diversity (calculated in thin section). (G) *P. sphaerica/P. ovalis* ratio. (H) Calcareous dinocysts assemblages based on changes of diversity and total number. (I) Foraminifera P/B ratio. (J) Distribution of particular morphotypes of planktonic foraminifera. (K) Abundance fluctuation of species and group of species of benthic foraminifera in foraminiferal assemblages. (L) Microfacies distinguished within section. (M) Relative sea-level curve for western Ukraine based on planktonic foraminifera. (N) Trophic conditions fluctuations.
(O) Paleoecological intervals. A, I–K, M after *Dubicka & Peryt (2012)*. Local planktonic foraminiferal zonation from *Walaszczyk & Peryt (1998)*. *H. helv.–Helvetoglobotruncana helvetica*.

obtained were cleaned in an ultrasonic bath and sieved into three fractions: <20 μm, 20–75 μm, and >75 μm. C-dinocyst specimens were handpicked from 20–75 μm fraction using a standard optical binocular microscope. Observations were performed using scanning electron microscopy (SEM; HITACHI 3-4700) at the Laboratory of Field Scanning Emission Microscopy and Microanalysis at the Institute of Geological Sciences of Jagiellonian University, Krakow, Poland.

Morphological characteristics such as shape, size, aperture, and wall ultrastructure are used for the taxonomic identification of c-dinocysts (*Nowak, 1963*; *Nowak, 1974*; *Bolli, 1974*; *Keupp, 1987*; *Reháková, 2000*; *Wendler, Gräfe & Willems, 2002*; *Wendler, Wendler & Huber, 2013*; *Omaña et al., 2014*). Four types of cyst wall ultrastructures, based on the crystal orientation within the wall, have been described: pithonellid, radial, oblique, and tangential (*Keupp, 1987*; *Kohring, 1993*; *Young et al., 1997*). The foraminiferal data of *Dubicka & Peryt (2012)* were plotted against the section with the c-dinocyst results obtained here.

The material used in this study is housed at the Pedagogical University of Krakow, Institute of Biology and Earth Sciences, Poland (collection no. UPKG/1/2022).

## RESULTS

### Microfacies

Five microfacies (Figs. 3L, 4) were distinguished within the Dubivtsi white chalk:

**MF1** Planktonic and benthic foraminiferal wackestone. This microfacies occurs in the lowest part of the Dubivtsi section (sample 1). It contains benthic and planktonic foraminifera (Fig. 4A) and rare bioclasts. No c-dinocysts is observed. The groundmass was mostly a micritic matrix, locally slightly recrystallized.

**MF2** Benthic foraminiferal wackestone-packstone (Figs. 4B and 4C). This microfacies occurs in samples 2 and 13. Grain components are represented by benthic and planktonic foraminifera (where the benthics dominate, reaching up to 80% of the foraminiferal assemblages), common bioclasts, and peloids (Fig. 4B). Only a few calcareous dinocysts represented by pithonellids are found. The components are embedded in a micritic matrix, locally slightly recrystallized. Locally burrows filled with coarsegrained materials are observed (Fig. 4C). Some roughly parallel streaks composed of chaotically, and tightly packed bioclasts are observed in sample 2.4

**MF3** Pithonellid-rich wackestone-packstone (Figs. 4D and 4E). This microfacies occurs in samples 3–10. Pithonellid floods, which can exceed 80% of grain components, are observed (Fig. 4D). Calcareous dinocysts are represented by two species only: *Pithonella ovalis* (Kaufmann in Heer 1865) Lorenz 1902 and *P. sphaerica* (Kaufmann in Heer 1865) (*Zügel, 1994*) (Assemblage 1; see section below). Small numbers of benthic and planktonic foraminifera are also observed. Benthic foraminifera dominated and locally exceeded 80% of the foraminiferal assemblages. Bioclasts are locally common. Peloids are common and very common locally (Fig. 4E). The groundmass is a dark gray micritic matrix.

**MF4** Calcareous dinocyst, foraminiferal wackestone-packstone (Figs. 4F and 4G). This microfacies occurs in the middle and upper parts of the section (samples from 17 to 25). It contained various c-dinocysts (Fig. 4F), represented by eight species (Assemblage 2; see section below). Benthic and planktonic foraminifera are also observed. Peloids and bioclasts are locally common (Fig. 4G). The components are embedded in a micrite matrix. Streaks composed of chaotically and tightly packed, more or less fragmented skeletal grains are locally visible (Fig. 4G).

**MF5** Calcareous dinocyst, foraminiferal wackestone (Fig. 4H). This microfacies occurs in the uppermost part of the section in samples 29–32. It contains a moderate number of c-dinocysts represented by four species (Assemblage 3; see section below). Benthic and planktonic foraminifera are also observed. Bioclasts are less prevalent. Peloids are common locally. Some orientation of the components, visible as streaks composed of chaotically, tightly packed larger skeletal elements, are locally observed. These components are embedded mainly in the micritic matrix.

### Vertical distribution of calcareous dinocysts

Calcareous dinocysts are present in fifteen of the sixteen samples (Supplemental Information 1, Figs. 5–8). Only one sample (no. 1) did not yield c-dinocysts (Fig. 3). The preservation of the c-dinocysts observed in the thin sections is generally good, even when the matrix is recrystallized. Locally, the specimens are poorly preserved therefore

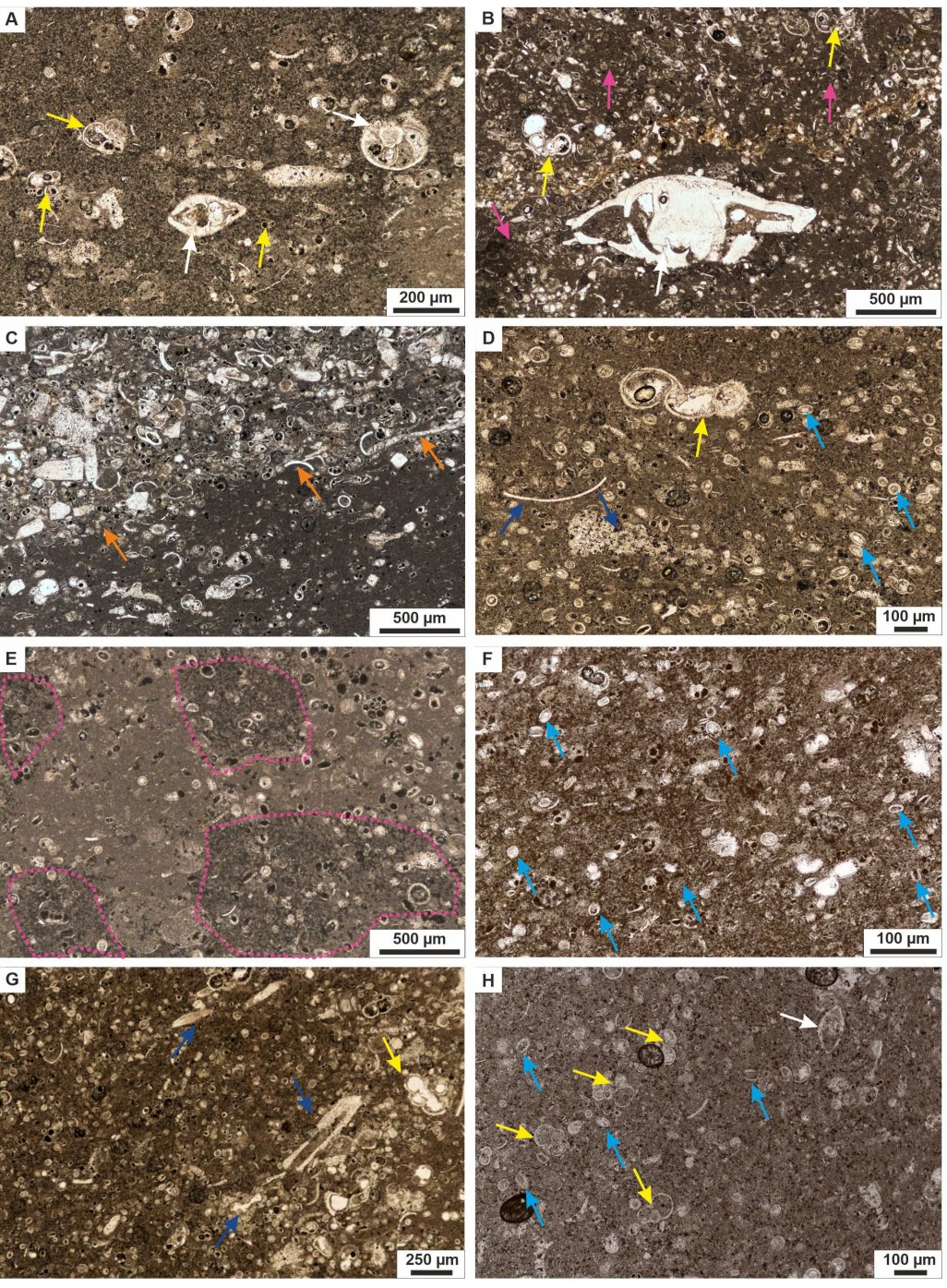

**Figure 4** **White chalk microfacies of the Dubivtsi section.** (A) MF1 Planktonic and benthic foraminiferal wackstone showing benthic foraminifera dispersed within micritic matrix (sample 1). (B, C) MF2 Benthic foraminiferal wackestone-packstone with benthic foraminifera (yellow arrow) and peloids (pink arrows) dispersed within micrite matrix (seen on B), and burrow filled with coarse-grained material and its contact (marked by orange arrow) with the fine-grained sediment (seen on C) (samples: B—13, C—2). (continued on next page...)

**Figure 4 (...continued)**
(D, E) MF3 Pithonellid-rich wackestone-packstone with large amount of pithonellids (oval and spherical cross sections marked by light blue arrows), rare foraminifera (yellow arrow), bioclasts (navy blue arrows), and peloids or chalk intraclasts (pink circles), (samples: D–4, E–8). (F, G) MF4 Calcareous dinocyst, foraminiferal wackestone-packstone, showing c-dinocysts (oval and spherical cross-sections) marked by light blue arrows, and foraminifera (yellow arrow) and common larger bioclasts (navy blue arrows); note a streak with the coarse-grained material in lower part of figure, (samples: F–20, G–17). (H) MF5 Calcareous dinocyst, foraminiferal wackestone with c-dinocysts (light blue arrows) and foraminifera, dispersed within micrite matrix (sample 32). Planktonic foraminifera are marked with yellow arrows while benthic foraminifera with white arrows. All thin-section photographs in plain polarized light.

detailed identification was not possible. These specimens were included as problematic taxa in the statistical analysis (Supplemental Information 1).

Only eight species belonging to three genera (*Pithonella*, *Bonetocardiella*, and *Stomiosphaerina*) were identified. *Pithonella* accounted for 93–100%, dominating all assemblages. The assemblages are predominantly represented by *Pithonella ovalis* (Kaufmann in Heer 1865) Lorenz 1902; *Pithonella sphaerica* (Kaufmann in Heer 1865) (*Zügel, 1994*) is less common; and *Pithonella lamellata* Keupp in Keupp & Kienel 1994 and *P. cardiiformis* (*Zügel, 1994*) are rare. *Bonetocardiella* represented by *Bonetocardiella conoidea* Bonnet 1956 appears in smaller numbers. The genus *Stomiosphaerina* represented by *Stomiosphaerina biedai* (*Nowak, 1974*) and *Stomiosphaerina bakae*, recently created from the material of the Dubivtsi section (*Ciurej, 2023*), also occurred in small numbers. Small numbers of unidentified specimens, namely Morphotype 1, were also detected. All species are illustrated in Figs. 5–8.

Most c-dinocyst species recognized are rather long-ranging. Biostratigraphically important is *S. biedai* (*Nowak, 1974*), which was described within Turonian - ?Santonian) deposits (*Nowak, 1974*). The other important species is *P. cardiiformis* Zügel, of which the first occurrence was described in the middle Cenomanian (*Zügel, 1994*) and the last occurrence in the upper Turonian (*Wiese, Zobel & Keupp, 2015*). Accordingly, the co-occurrence of *S. biedai* and *P. cardiiformis* indicates the upper Turonian.

Species diversity and the number of specimens that varied among individual samples were calculated. The three c-dinocyst assemblages were distinguished based on the vertical profile distribution (Supplemental Information 1). These are as follows:

**Assemblage 1** observed within samples 2–13 (Fig. 3H), comprises a strongly depleted assemblage, represented by only two species, *P. ovalis* and *P. sphaerica*. *P. ovalis* prevailed in all samples. Three sub-assemblages were distinguished based on the number of specimens.

Sub-assemblage 1A, observed in sample 2, is characterized by a negligible number of 37 specimens of pithonellids, including 27 of *P. ovalis* and 8 of *P. sphaerica* (*P. ovalis* accounts for 77% and *P. sphaerica* for 23% of the assemblage) (Figs. 3B, 3C).

Sub-assemblage 1B, occurs in samples 3–10, and is characterized by a flood abundance of pithonellids; the pithonellids varied from 1,480 to 2,656 specimens, including 1,045 to 2,041 specimens of *P. ovalis* and 375 to 599 specimens of *P. sphaerica* (*P. ovalis* constitutes from 70 to 84% and *P. sphaerica* constitutes from 16% to 30% of the assemblage). *P. sphaerica* shows a decreasing trend in younger samples.

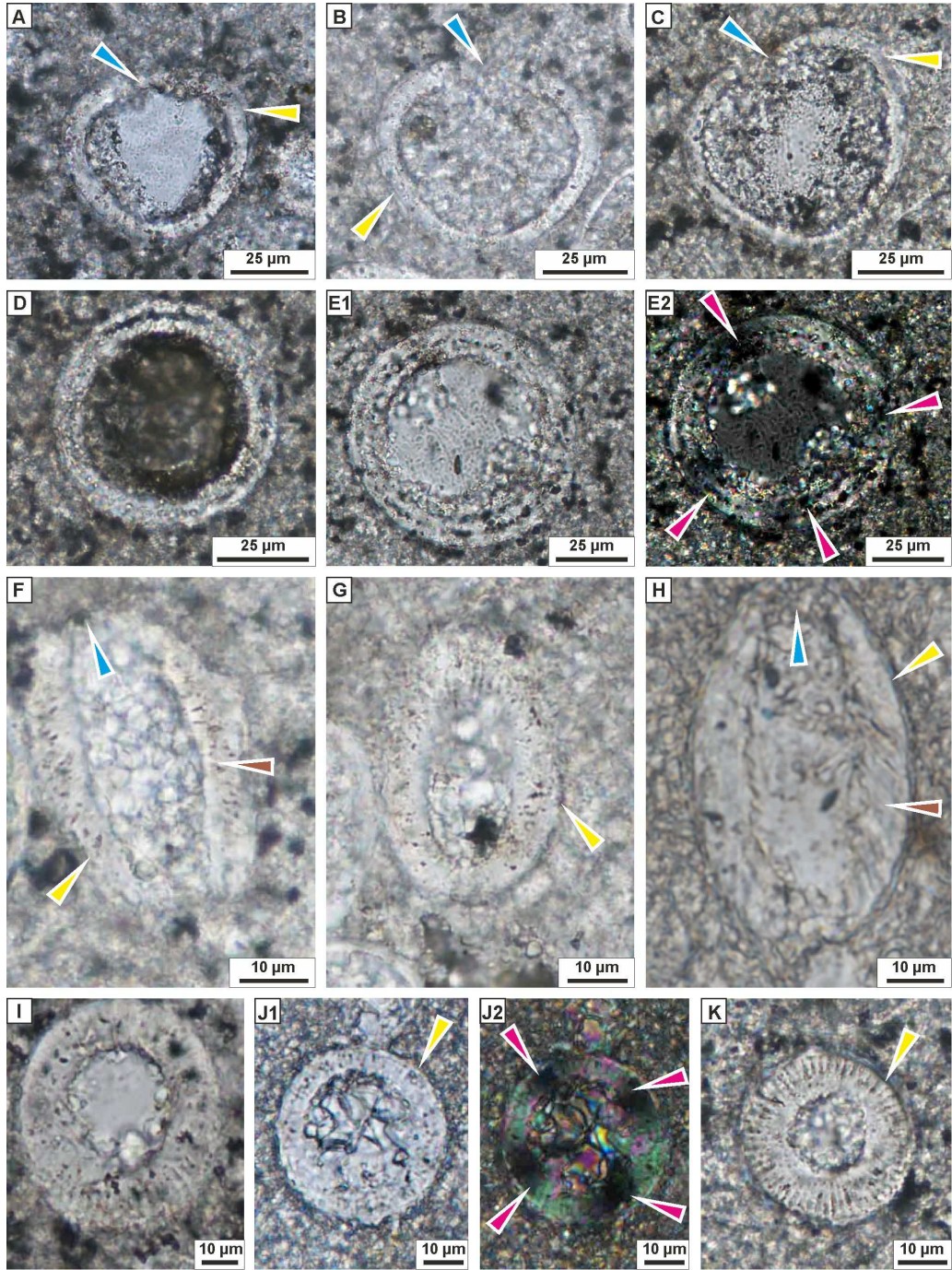

**Figure 5** **Transmitted light microscopy images of calcareous dinocysts from the Dubivtsi section.** (A–C) *Bonetocardiella conoidea* Bonnet 1956 showing the pithonellid wall (yellow arrows) and aperture (blue arrows), (sample 17). (D, E) *Pithonella lamellata* Keup in Keupp and Kienel 1994 showing wall with the complexlayers , and extinction cross (purple arrows), (sample 17). (F–K) *Pithonella ovalis* (Kaufmann in Heer 1865) Lorenz 1902, (F, G, I–sample 4; H–sample 7, J, K–sample 17) showing the thick wall with double layer (yellow arrows), and the pithonellid wall type (brown arrows), the aperture (blue arrows) and extinction cross (purple arrows): (F–H) longitudinal sections; (I–K) axial sections. All images were taken from thin sections under plain-polarized light, except E2 and J2 which are under crossed polars.

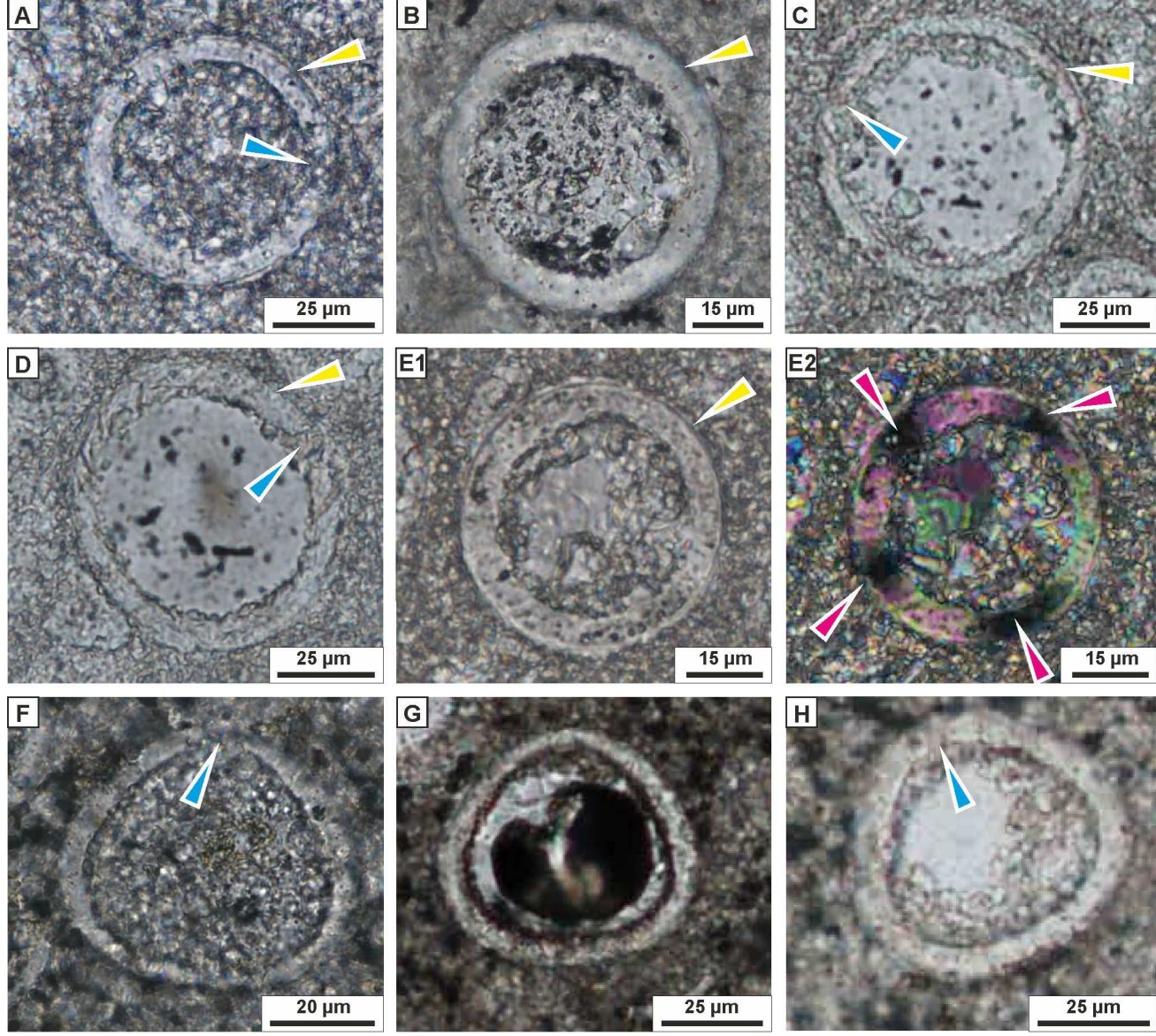

**Figure 6** **Transmitted light microscopy images of calcareous dinocysts from the Dubivtsi section.** (A–E) *Pithonella sphaerica* (Kaufmann in Heer 1865) *Zügel, 1994*; showing the thick wall with double layer (yellow arrows), and the aperture (blue arrows) and extinction cross (purple arrows), (samples: A–3, B–4, C–E–7). (F–H) *Pithonella cardiiformis* *Zügel, 1994*, with aperture (blue arrows), (sample 17). All images were taken from thin sections under plain-polarized light, except E2 that was under crossed polars.

Sub-assemblage 1C, observed in sample 13, is marked by a significant decrease in c-dinocysts: the number of pithonellids dropped to 174 specimens, with 123 specimens of *P. ovalis* and 46 specimens of *P. sphaerica* (*P. ovalis* accounts for 73% and *P. sphaerica* of 27% of assemblage).

**Assemblage 2** is characterized by an increase in c-dinocyst diversity but a decrease in numbers. Eight species, namely *P. ovalis*, *P. sphaerica*, *P. lamellata*, *P. cardiiformis*, *B. conoidea*, *S. bakae*, *S. biedai*, and Morphotype 1, were recognized. This assemblage

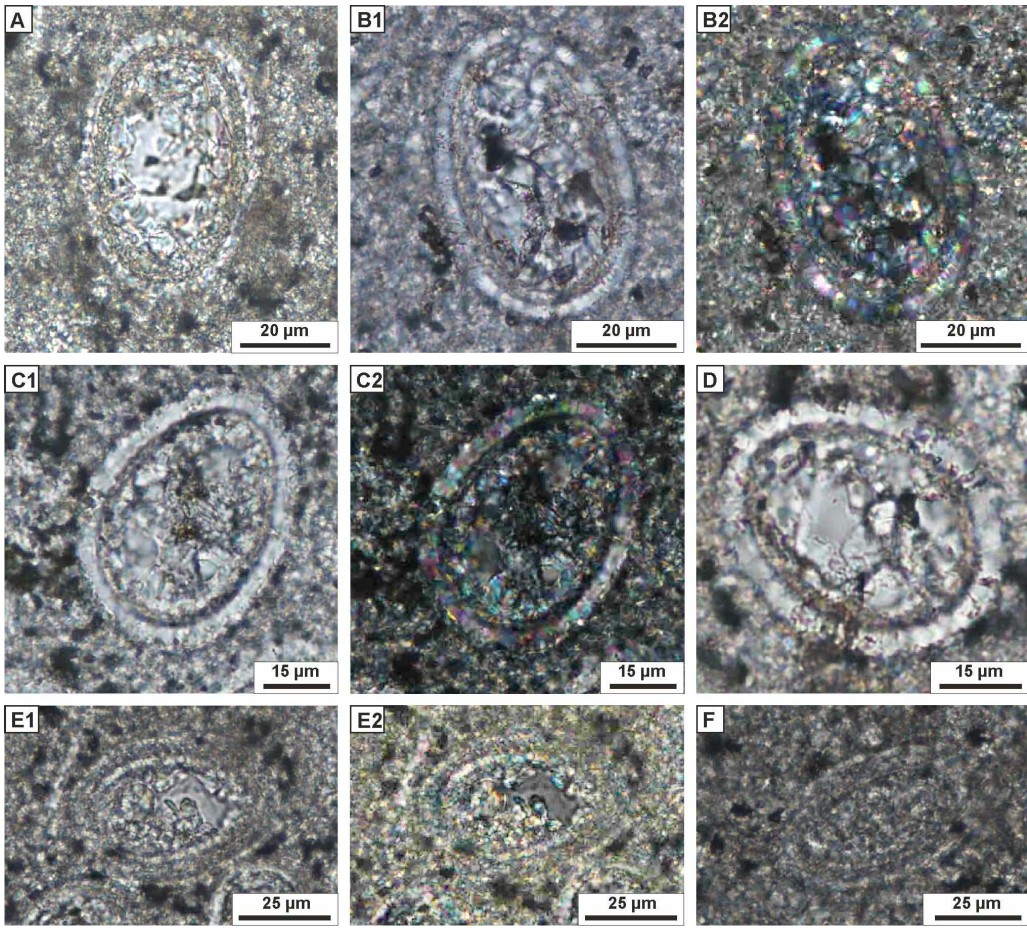

**Figure 7** **Transmitted light microscopy images of calcareous dinocysts from the Dubivtsi section.** (A, B) *Stomiosphaerina biedai* (*Nowak, 1974*), (samples: A–20; B–22). (C, D) *Stomiosphaerina bakae* Ciurej 2023, (sample 17). (E), (F) Morphotype 1 (samples: E–17, F–32). Thin sections, plain-polarized light, except B2, C2, E2–crossed polars.

occurs in samples 17–25 (Fig. 3H). A very high number of c-dinocysts, at 1,568 specimens, was observed in sample 17, followed by a marked decrease to 652 specimens in sample 20, and a gradual decrease to 484 specimens in sample 25. The most abundant species within Assemblage 2 is *P. ovalis* (ranging from 296 to 1,118 specimens (which is 60 to 72% of assemblage in samples), and *P. sphaerica* is less abundant: from 134 to 337 specimens, and constitutes of 21 to 30% of assemblage). Other pithonellids include from 8 to 39 species, and not exceed 3% of assemblage in samples. Other c-dinocysts range from 31 to 59 specimens, and accounts for <7% of the assemblage (Figs. 3B–3D).

**Assemblage 3** is characterized by a decrease in c-dinocyst diversity of 50% compared with Assemblage 2. Four species observed herein were represented by *P. ovalis, P. sphaerica, B. conoidea,* and Morphotype 1, all known from previous assemblages. The number of specimens ranged from 454 to 403. The dominant species, *P. ovalis,* ranged from 329 to 411, and constitutes from 82 to 92% of the assemblage. The second species, *P. sphaerica,*

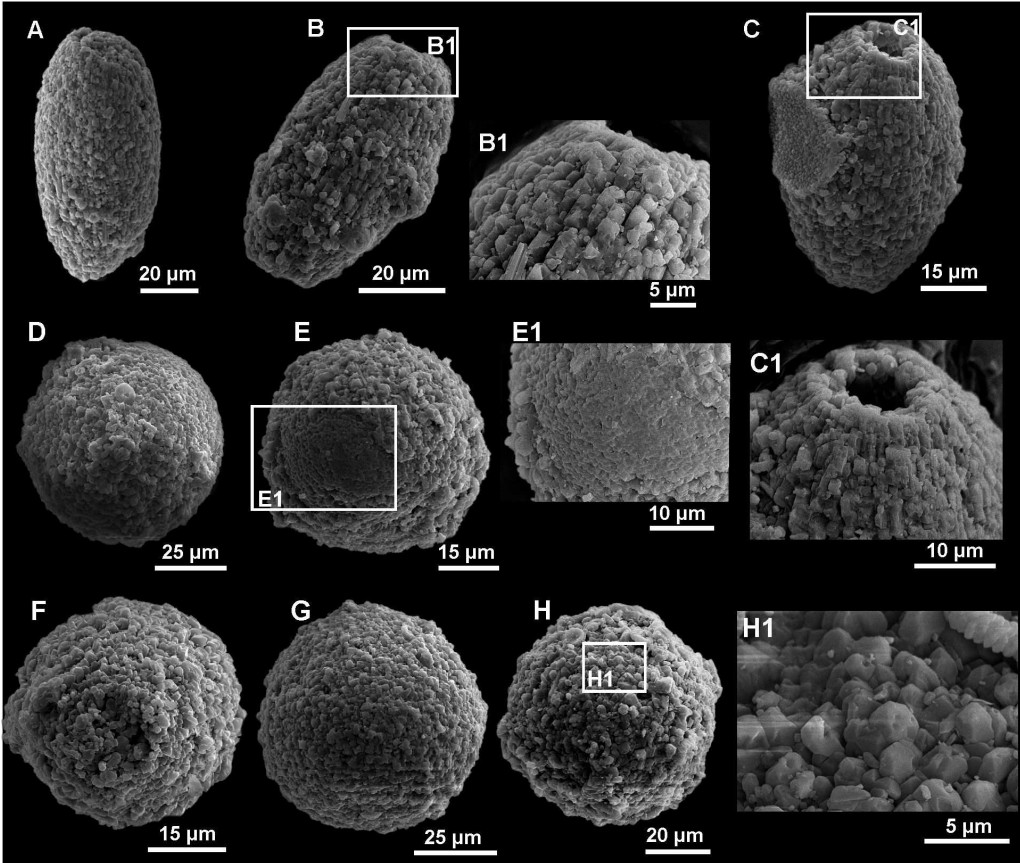

**Figure 8** **SEM images of calcareous dinocyst from the Dubivtsi section.** (A –C) *Pithonella ovalis* (Kaufmann in Heer 1865) Lorenz 1902: (A) Longitudinal view of *P. ovalis* showing outer wall with diagenetic (syntaxial) calcite crystals overgrowths (sample 2). (B) Longitudinal view of *P. ovalis* showing outer wall texture (sample 3). (B1) Magnified view of the same specimens (sample 3). (C) Longitudinal view of *P. ovalis* of outer wall with blocky calcite structures (sample 6). (C1) Magnified view of the same specimens showing details of aperture (sample 6). (D–H) *Pithonella sphaerica* (Kaufmann in Heer 1865) *Zügel, 1994* in various stage of preservation: (D) Outer wall with blocky, secondary calcite structure (sample 3). (E) Specimen with partially removed thin outer wall (sample 4). (E1) Close-up of the same specimens with details of inner wall (sample 4). (F, G) Variously diagenetically altered outer wall where the blocky, recrystallized calcite structure is visible (F—sample 3, G—sample 6). (H, H1) outer wall with blocky, recrystallized calcite structure (sample 7).

ranged from 31 to 54 specimens, and constitutes from 8 to 13% of the assemblage. It shows a clear downward trend toward the younger sediments. Other species (represented only by *B. conoidea* and Morphotype 1) contain from 14 to 18 specimens, and constitutes up to 5% of assemblage (Figs. 3B–3D). This assemblage occurs in the highest part of the section (samples 29–32) (Fig. 3H).

## Paleoecological intervals

Based on the assemblages of c-dinocysts and planktonic and benthic foraminifera, together with microfacies analysis, four paleoecological intervals were distinguished (Fig. 3-O) as follows:

**Interval 1** is characterized by the absence of calcareous dinocysts (sample 1). Planktonic and benthic foraminifera are relatively diverse and abundant. Planktonic foraminifera in samples from this interval represent almost 50% of the assemblage and are composed of deep- and shallow-dwelling taxa belonging to the following genera: *Helvetoglobotruncana, Marginotruncana, Dicarinella, Whiteinella, Falsotruncana, Globigerinelloides, Heterohelix,* and *Hedbergella.* Benthic foraminifera are dominated by *Gavelinella, Gyroidinoides, Lenticulina, Globorotalites, Marssonella, Arenobulimina,* and *Valvulineria.* This interval corresponds to microfacies MF1, planktonic and benthic foraminiferal wackestone (Fig. 3L).

**Interval 2** correlates to c-dinocysts Assemblage 1 (samples 2–13) including a sub-assemblage of very low species diversity, composed of only two species, and a very high number of c-dinocysts (samples 3–10) and a sub-assemblage with a much lower number of specimens (samples 2 and 13). The planktonic foraminiferal assemblage in samples from this interval differs from that of Interval 1 by a drastic decline in the number of planktonic foraminifera and a disappearance of deep-dwelling forms (*Helvetoglobotruncana, Marginotruncana, Dicarinella,* and *Falsotruncana*). Benthic foraminiferal assemblages also differ from those of Interval 1, primarily due to the disappearance of representatives of the genus *Valvulineria.* The benthic foraminifera of the upper part of Interval 2 have very low diversity and are composed of only five genera: *Gavelinella, Gyroidinoides, Lenticulina, Globorotalites,* and *Arenobulimina.* Interval 2 correlates with two different microfacies: MF2, benthic foraminiferal wackestone-packstone microfacies (samples 2 and 13) (Fig. 3L) and MF3, pithonellid-rich wackestone-packstone (samples 3–10) (Fig. 3L).

**Interval 3** is characterized by c-dinocysts Assemblage 2 (samples 17–25) that is moderately diverse (eight species) with an intermediate number of c-dinocyst specimens. Planktonic foraminiferal assemblages in samples from this interval are similar to those of Interval 2, whereas benthic foraminiferal assemblages differ mainly by the occurrence of deep-infaunal foraminifera *Praebulimina.* This interval corresponds to the MF4, c-dinocyst, foraminiferal wackestone-packstone microfacies (Fig. 3L).

**Interval 4** correlates to c-dinocysts Assemblage 3 (samples 29–32) characterized by low diversity (four species) and a moderate number of c-dinocysts. Planktonic foraminiferal assemblages in samples from this interval differ from that of Intervals 1 and 2 by the re-appearance of deep-dwelling forms (*Marginotruncana* and *Dicarinella*), while benthic foraminiferal assemblages differ by the disappearance of *Praebulimina* and the occurrence of *Eponides.* This interval is correlated with the MF5, c-dinocyst, foraminiferal wackestone microfacies (Fig. 3L).

## DISCUSSION

### Calcareous dinocyst distribution in relation to the water depth

Two species of the genus *Pitonella* (*P. ovalis* and *P. sphaerica*) are the main components of the assemblages in the Dubivtsi section, with evident dominance of *Pithonella ovalis.* The *P. sphaerica* to *P. ovalis* ratio (*Ps/Po* ratio) was introduced by *Keupp (1991)* and used by *Zügel (1994),* and *Wendler, Gräfe & Willems (2002)*) to assess the water depth of the

basin and distance from the shoreline. According to these authors, a high *Ps/Po* ratio ∼10 characterizes shelf assemblages with a coastal influence. A low *Ps/Po* ratio ∼3 suggests that the assemblages originated from an outer shelf/distal environment. Throughout the entire Dubivtsi section, the *Ps/Po* ratio was very low, ranging from 0.42 to 0.08. This ratio suggests that the Dubivtsi chalk was deposited in an outer shelf setting. However, the question arises as to how to interpret the inner and outer shelves of the Cretaceous. A continental shelf is part of a continent that is submerged under seawater. Following this definition, the entire Cretaceous epicontinental sea should be interpreted as a shelf, but today's continental shelves are narrow and largely restricted to continental margins. Therefore, a direct link between the present and the 'non-actualistic' Cretaceous marine depositional environment is difficult to establish. Nonetheless, we interpret that the low Ps/Po ratio in the Dubivtsi section likely indicates instead a significant distance from an emerged area.

The Dubivtsi chalk was deposited in a rather shallow epicontinental sea that was influenced by sea-level fluctuations. Based on planktonic foraminiferal water depth indices (see *Bé & Hamlin, 1967*; *Bé, 1977*; *Hart & Bailey, 1979*; *Caron & Homewood, 1983*; *Leckie, 1987*), it can be concluded that the water depth during deposition of Intervals 2 and 3 was ca. 50–100 m while during intervals 1 and 4 it was greater than 100 m (*Dubicka & Peryt, 2012*). The sea level fall during Interval 2 is believed to be related to a global eustatic sea-level fall (Fig. 9) during early and middle part of the late Turonian (see *Wiese & Kröger, 1998*; *Wiese et al., 2004*; *Jarvis et al., 2015*). However, the influence of local tectonic processes cannot be excluded.

The sea level curve interpreted from foraminiferal data correlates with recent studies of c-dinocyst assemblages and also with the microfacies analysis (see Figs. 3H–3M). A significant sea level fall indicated in the lowest part of the section (between samples 1 and 2) correlates with the appearance of c-dinocysts in the chalk deposits, though still with negligible numbers of specimens (37 specimens) and with very low species diversity (2 species) (Sub-assemblage 1A). The low sea level occurred during Interval 2, which corresponds to a flood abundance (up to 2,656 specimens) but very low diversity (2 species) Sub-assemblage 1B. The transgressive episode likely started within Interval 3, which correlates with an increase in c-dinocyst diversity (eight species) but a decrease in their numbers (up to 1,568 specimens) (Assemblage 2). A significant sea level rise observed in the uppermost part of the section (Interval 4), corresponds to low abundance (up to 454 specimens) and diversity (four species) c-dinocyst Assemblage 3. The microfacies from Interval 4 also suggests a transgressive event, as bioclasts are less common.

### Nutrient availability and evolution of the basin

The study of the Ukrainian chalk revealed a relatively shallow water setting c.a. between 50 and 100 m depth during Intervals 2 and 3 (Fig. 2-O) which were, however, enhanced by a small amount of terrigenous influx from the land area. The minimal terrigenous influx into the basin is indicated by nearly pure calcareous facies, with little or no detrital content, such as quartz or clay minerals (*Dubicka, Peryt & Szuszkiewicz, 2014*). The Dubivtsi chalk has an exceptionally high $CaCO_3$ content, ranging from 97.8% to 99.9%. Additionally, the chalk displays diamagnetic properties that are characterized by a very low negative

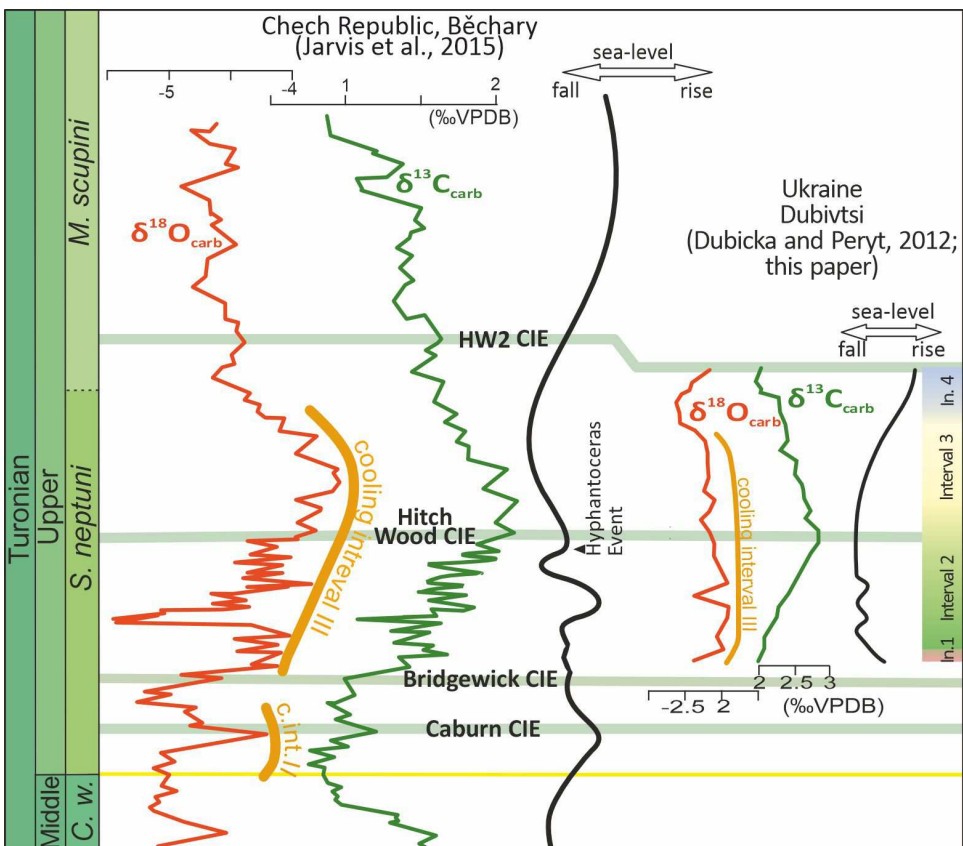

**Figure 9  Stable carbon and oxygen isotope profiles compared to an inferred sea-level curves for Dubivtsi and the Bohemian Cretaceous Basin.** Data for the Bohemian basin after *Uličný et al. (2014)* and *Jarvis et al. (2015)*. Late Turonian cooling intervals II and III after *Voigt & Wiese (2000)*. Intervals 1–4 are paleoecological intervals based on the assemblages of c-dinocysts and planktonic and benthic foraminifera (this study). CIE–carbon isotope event.

value of magnetic susceptibility ($\chi$), with a mean value of $-0.5 \times 10^{-8}$ m³ kg⁻¹ (*Dubicka, Peryt & Szuszkiewicz, 2014*). Individual samples exhibited slightly increased values up to $0.8 \times 10^{-8}$ m³ kg⁻¹ (sample 15). This phenomenon is likely due to the basin's paleogeographic position and its relatively large distance from the shoreline (Fig. 1A). During sedimentation of Intervals 1–3 the nearest land area might be associated with the Ukrainian Shield (*Pasternak et al., 1968*; *Gavrilishin, Pasternak & Rozumeyko, 1991*, and Fig. 1A), which was situated 200–300 km to the east.

The c-dinocyst assemblage in the studied Dubivtsi section was characterized by low species richness dominated by *Pithonella*, with some shifts from two species occurring in the lower part (Assemblage 1) to eight taxa in the middle part (Assemblage 2) and four species in the highest part (Assemblage 3). A very low-diverse with only seven species of c-dinocyst assemblage, and dominated by *Pithonella* has also been observed in the upper Turonian of the Lower Saxony Basin, northern Germany (*Wiese, Zobel & Keupp, 2015*). Such a low species richness was interpreted as a 'nutrient crisis caused by the

drowning of source areas of nutrient input during the early Late Cretaceous sea-level highstand' (*Wiese, Zobel & Keupp, 2015*, p.1). Furthermore, *Gale et al. (2000)* linked a high abundance of c-dinocysts in southern England with a reduction in water productivity during the Cenomanian–Turonian boundary interval (see also *Dubicka et al., 2021*). A spread of calcareous nannoplankton/planktonic foraminifera/c-dinocyst-rich sediments of the Cenomanian and Turonian has been previously recognized (*Hart, 1991*; *Hay, 2008*; *Pearce, Jarvis & Tocher, 2009*; *Linnert, Mutterlose & Erbacher, 2010*) from England to the Russian Platform. In general, it is interpreted that during the Turonian the highly oligotrophic system of the European shelf sea was established (*Haq, 2014*; *Gale et al., 2000*; *Pearce, Jarvis & Tocher, 2009*; *Linnert, Mutterlose & Erbacher, 2010*).

Considering the above, the highly abundant but low-diversity c-dinocyst Assemblage 1 (Interval 2), apparently resulted from reduced surface water productivity and an oligotrophic environment, which is additionally supported by data from foraminiferal assemblages (Figs. 3J, 3K, 3M). Oligotrophic conditions of Interval 2, coinciding with the period with lower relative sea level recognized in western Europe (*Wiese et al., 2004*; *Jarvis et al., 2015*), may appear surprising. Usually, regression is related to less oligotrophic conditions as the sea retreat exposes land areas, affecting winnowing increase and a higher input of nutrients from the land to the sea. However, the large distance between Dubivtsi and land areas (Fig. 1A) likely limited the supply of nutrients from hinterland sources even during sea-level fall. On the other hand, shelf upwelling (*Merino, 1997*; *Zaytsev et al., 2003*; *Jing et al., 2009*) from the nearby Carpathian Basin (today the Carpathian Orogen is located ca. 50 km to the south), that likely injected nutrients into surface waters during sea-level highstands, might have been halted during subsequent lowstands.

Assemblage 2 which is characteristic of Interval 3 is likely related to more mesotrophic conditions within the photic zone, which could have favored an increase in species richness to eight species (six new species of c-dinocysts plus *P. ovalis* and *P. sphaerica*). The higher trophic level is also supported by the significant increase in the deep-infaunal benthic foraminifera *Praebulimina* (Fig. 3K). During Interval 3, the studied area was still under a relatively low sea level, however, progressive deepening may have had greater impact on nutrient input from the Carpathian Basin. In addition, nutrients may have been supplied by runoff from the adjacent land in the Stryi region, named ''Krukienic Island'' (*Pasternak, 1959*), located some 40–50 km to the west Dubivtsi. The island was uplifted during the Subhercynian tectonic phase that commenced in the late Turonian (*Dubicka, Peryt & Szuszkiewicz, 2014*) and caused extensive basin inversion in Central Europe (see *Walaszczyk, 1992*; *Vejbæk & Andersen, 2002*; *Voigt, Kley & Voigt, 2021*). The uplift of this area close to the European continental margin might be supported by the increase in mass magnetic susceptibility and the decrease in $CaCO_3$ content (*Dubicka, Peryt & Szuszkiewicz, 2014*).

A slight decrease in nutrients under mesotrophic/oligotrophic conditions (Interval 4) may have influenced c-dinocyst Assemblage 3 (Fig. 3H). This is interpreted by a 50% reduction of c-dinocyst species diversity in relation to Interval 3 (Fig. 3O) and the disappearance of the infaunal foraminifera *Praebulimina* (Figs. 3F, 3K). The renewed drop of the trophic conditions could have been related to the inundation of the emerging

areas during transgression as shown by the reappearance of deep-dwelling planktonic foraminifera. This theory was confirmed by the shift in the negative direction of the mass magnetic susceptibility of the samples from this interval (*Dubicka, Peryt & Szuszkiewicz, 2014*).

The study interval of the Late Turonian was associated with evidence of significant climate cooling (Cooling Interval III; Fig. 9) throughout Europe (*Voigt & Wiese, 2000*; *Jarvis et al., 2015*). Because c-dinocyst Assemblages 1–3 correlate with rather stable $\delta^{18}$O values, climate variability is not regarded as a key factor driving recorded c-dinocyst assemblage changes. Nevertheless, organic walled dinocysts display very significant changes in adjacent basins during this time interval (*e.g.*, *Olde et al., 2015*).

In summary, the data from this study show that nutrient availability gradients, which are a function of water depth and distance from the shore, strongly determined changes in the calcareous dinocyst assemblages.

## ACKNOWLEDGEMENTS

The authors are grateful to Danuta Peryt (Warsaw University, Warsaw, Poland) for providing samples for this research and helpful remarks. We thank Daniela Reháková for comments on some of the systematics. We appreciate helpful reviews by Ian Jarvis, Malcolm Hart and an anonymous reviewer. Special thanks go to Michael Kaminski for improving the English text.

### Funding

This work was supported by the Faculty of Exact and Natural Sciences, Pedagogical University of Krakow, Poland, Statutory Funds, Project no. WPBU/2022/04/00194 and the National Science Centre, Poland, grant no. 2017/27/B/ST10/00687. The funders had no role in study design, data collection and analysis, decision to publish, or preparation of the manuscript.

### Grant Disclosures

The following grant information was disclosed by the authors:
Faculty of Exact and Natural Sciences, Pedagogical University of Krakow, Poland, Statutory Funds: WPBU/2022/04/00194.
National Science Centre, Poland: 2017/27/B/ST10/00687.

### Competing Interests

The authors declare there are no competing interests.

### Author Contributions

- Agnieszka Ciurej conceived and designed the experiments, performed the experiments, analyzed the data, prepared figures and/or tables, authored or reviewed drafts of the article, and approved the final draft.

- Zofia Dubicka conceived and designed the experiments, prepared figures and/or tables, authored or reviewed drafts of the article, and approved the final draft.
- Andriy Poberezhskyy conceived and designed the experiments, authored or reviewed drafts of the article, and approved the final draft.

## Data Availability

The raw measurements are available in the Supplementary File.

## Supplemental Information

Supplemental information for this article can be found online at http://dx.doi.org/10.7717/peerj.16201#supplemental-information.

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
