# Peer review of "Calcareous dinoflagellate blooms during the Late Cretaceous ‘greenhouse’ world—a case study from western Ukraine"

_PeerJ, doi:10.7717/peerj.16201_

## Round 0.1 · original submission · Minor Revisions

Please, address all the reviewers' comments. Notice that two reviews contain annotated manuscript with additional comments and corrections.

·

Basic reporting

Clearly written study placed in context with appropriate literature quoted. Well-structured with raw data provided in Supplementary table. Some improvement in English style and terminology would be beneficial – suggestions for this have been included in an annotated pdf.
I would recommend use of UK English rather than American English throughout. This paper describes European not North American rocks.

Experimental design

Content appropriate for journal. Clearly defined research question addressed using well-established analytical methods with appropriate description.

Validity of the findings

All data are provided with clear conclusions. However, these rely very heavily on a previous foraminiferal study of the study section and do not fully address the wider European to global context.

Additional comments

This regional study documents changes in the micropalaeontology and microfacies through an upper Turonian succession in western Ukraine. The study and discussion have broader implications for understanding Turonian environmental change in Europe during a key interval of the Late Cretaceous. The original data presented are relatively limited but will be of wide interest and warrant publication.
The paper makes appropriate use of previously published carbon stable isotope and foraminiferal biostratigraphic data to place the c-dinocyst results in context. However, there are aspects that might be addressed further.
(1) The full stratigraphic extent of the study section is rather poorly constrained. The carbon isotope stratigraphy and micropalaeontology should enable the authors to more clearly place the bottom and top of the section within the context of the complete Turonian succession described in the Czech Republic and beyond, see e.g. Jarvis et al. (2015), Plint et al. (2022) and references therein.
(2) The attribution of the changes in the study section to eustatic sea-level change is poorly justified, with passing reference to the Hyphantoceras Event. The eustatic versus relative sea-level history of the Late Turonian is complex and relatively well constrained with respect to both biostratigraphy and carbon isotope stratigraphy. The authors should consider e.g. Jarvis et al. (2015), Plint et al. (2022) and references therein, and place the study more specifically in a wider context.
(3) The study interval of the Late Turonian was associated with evidence of significant climate cooling and palaeoceanographic changes throughout Europe that are not mentioned. These need to be considered even if their impact is not regarded as being significant for the c-dinocyst story, although this would need to be justified. Organic walled dinocysts display very significant changes in adjacent basins during this time interval (e.g. Olde et al. 2015).
(4) I have raised some issues with the microfacies descriptions and interpretation that the authors might consider.

Some additional references for consideration:
Gradstein FM, Ogg JG, Schmitz MD and Ogg GM, eds. (2020) Geologic Time Scale 2020. Amsterdam: Elsevier. doi: 10.1016/C2020-1-02369-3
Hart MB (1991) The Late Cenomanian calcisphere global bioevent. Proceedings of the Ussher Society 7, 413-17.
Jarvis I, Trabucho-Alexandre J, Gröcke DR, Uličný D and Laurin J (2015) Intercontinental correlation of organic carbon and carbonate stable isotope records: evidence of climate and sea-level change during the Turonian (Cretaceous). The Depositional Record 1, 53-90.
Laurin J, Čech S, Uličný D, Štaffen Z and Svobodová M (2014) Astrochronology of the Late Turonian: implications for the behavior of the carbon cycle at the demise of peak greenhouse. Earth and Planetary Science Letters 394, 254-69.
Olde K, Jarvis I, Pearce MA, Uličný D, Tocher BA, Trabucho-Alexandre J and Gröcke D (2015) A revised northern European Turonian (Upper Cretaceous) dinoflagellate cyst biostratigraphy: integrating palynology and carbon isotope events. Review of Palaeobotany and Palynology 213, 1-16.
Plint AG, Uličný D, Čech S, Walaszczyk I, Gröcke DR, Laurin J, Shank JA and Jarvis I (2022) Trans-Atlantic correlation of Late Cretaceous high-frequency sea-level cycles. Earth and Planetary Science Letters 578, 1-11.

Reviewer 2 ·

Basic reporting

English is good with some minor changes, noted on line 27, and some needed clarifications on line 153 check on the annotated pdf.
References are very good and complete.
in my opinion, upper Turonian should be Upper Turonian, however, I know is becoming common to see it as lowercase.

Fig. 1C is missing but mentioned in the text (lines 135, 140, 142).
Line 220 (Supplementary 1, fig. 4-7) is not on the package.
Line 235 "Unidentified specimens, namely Species A" should be named as ICZN indicates. also is not illustrated as the following line 236 states.
In lines 339, 340, 345 367, and 380 the word Diversity modify by low, very low should be defined better or changed to actual numbers/percentages.

Experimental design

no comment

Validity of the findings

The point about Late Cretaceous bloom in the area and the possible connections to other places is well made, if so clarified on the points above mentioned.

Annotated reviews are not available for download in order to protect the identity of reviewers who chose to remain anonymous.

·

Basic reporting

See below

Experimental design

See below

Validity of the findings

See below

Additional comments

The Abstract is OK, though I do feel that there should be a definition of acme (numbers, diversity, etc.) to make things clearer for the readers.

The Keywords are often rather carelessly selected in many papers and, in this case, they should think of the order (location, age, fossil groups, etc. – begin with bigger picture and focus inwards)

Line 57 – repeat of acme which is still not defined in terms of numbers or diversity.

Seawater chemistry is covered in numerous papers (including Van Dijk et al., 2016) but few of these are mentioned.

If these be calcareous dinocysts (which I agree with) then there should be mention of paratabulation (lines 70-87) as described by Wendler et al. and also Leighton et al., 2011, Cretaceous Research. In Jarvis et al. (1988, Cretaceous Research) there was a discussion of the replacement of organic-walled dincysts by calcareous forms associated with OAE2 (also reported in Simmons at this level in the Oman Mountains).

Line 148 – only 16 samples!

Lines 150-161 – methods of counting could be better explained.

In the discussion of assemblages there is no mention of ages in the text, which would help readers.
The authors use Gavelinella but some of these taxa might well have been revised by Stefan Revets in 2001 (Cushman Foundation).

Lines 330-334 – depth estimates look on the low side as, at 50 m, algal growth would have been an issue and this is not described. In the North Sea Basin concentrations of pithonellids in the Gorm Oil-field seem related th fault blocks and the upwelling caused by such sea floor features.

Figure 1 is really very vague and there are better maps available (e.g., Shell atlas and also the palaeogeographical atlas published by the Geological Society).

The sea level curves in Figure 2 are very vague and could be improved.

In the thin-sections arrows point to foraminifera but in a few cases the determination could be better – at least to generic level (even though this probably does not matter, it does assure readers that the authors know their taxa).

Interesting paper with much to commend it and some of the above should be readily corrected and the paper improved.

---

## Round 0.2 · accepted · Accept

Thank you for accommodating reviewers' suggestions and comments.